# Mesenchymal Stem Cell Transplantation Has a Regenerative Effect in Ischemic Myocardium: An Experimental Rat Model Evaluated by SPECT-CT Assessment

**DOI:** 10.3390/diagnostics14040401

**Published:** 2024-02-12

**Authors:** Antonella Koutela, George Loudos, Maritina Rouchota, Dimitrios Kletsas, Andreas Karameris, George Vilaras, George C. Zografos, Despoina Myoteri, Dimitrios Dougenis, Apostolos E. Papalois

**Affiliations:** 11st Department of Propaedeutic Surgery, Hippokratio Hospital, National and Kapodistrian University of Athens, 11527 Athens, Greece; gzografo@med.uoa.gr (G.C.Z.); apostolospapalois@gmail.com (A.E.P.); 2Department of Cardiac Surgery, Attikon University Hospital, Medical School, National and Kapodistrian University of Athens, 11527 Athens, Greece; ddougen@gmail.com; 3Department of Thoracic Surgery, Red Cross Hospital, 11527 Athens, Greece; 4Experimental, Educational and Research Centre ELPEN, ELPEN, 11527 Athens, Greece; 5BIOMTECH Laboratories, 15341 Athens, Greece; george@bioemtech.com (G.L.); mrouchota@bioemtech.com (M.R.); 6Laboratory of Cell Proliferation and Ageing, Institute of Biosciences and Applications, NCSR “Demokritos”, 15341 Athens, Greece; dkletsas@bio.demokritos.gr; 7Department of Pathology, NIMTS Hospital, 11521 Athens, Greece; akarameris@gmail.com (A.K.); vilarasgeorge@gmail.com (G.V.); 8Department of Pathology, Aretaieion University Hospital, Medical School, National and Kapodistrian University of Athens, 11528 Athens, Greece; dmyoteri@med.uoa.gr; 92nd Department of Surgery, Aretaieion University Hospital, Medical School, National and Kapodistrian University of Athens, 11528 Athens, Greece

**Keywords:** myocardial regeneration, stem cell transplantation, experimental, immunohistochemistry, SPECT imaging

## Abstract

Translational perspective: Ischemic heart disease remains a major medical problem with high mortality rates. Beside the great efforts devoted to research worldwide and the use of numerous experimental models, an absolute understanding of myocardial infarction and tissue loss has not yet been achieved. Furthermore, the regeneration of myocardial tissue and the improvement of myocardial activity after ischemia is one of the major areas of interest in the medical (and especially cardiovascular) community. In a novel experimental rat model, the beneficial effect of mesenchymal stem cell transplantation (MSCT) in a surgically induced ischemic myocardium was documented. From a clinical perspective, this work supports the surgical administration of MSCT in the infarcted area during coronary artery bypass surgery. Aims: The regeneration of myocardial tissue and the improvement of myocardial activity after ischemia is one of the major areas of interest in cardiovascular research. We developed a novel experimental rat model and used it to examine the effect of mesenchymal stem cell transplantation (MSCT) on myocardial ischemia evaluated by SPECT-CT and immunohistochemistry. Methods and results: An open thoracotomy took place for forty adult female Wistar rats with (*n* = 30) or without (*n* = 10) surgical ligation of the left anterior descending coronary artery (LAD) in order to cause myocardial ischemia. Myocardial viability was evaluated via SPECT/CT 7 days before surgery, as well as at 7 and 14 days post-surgery. At day 0, 15 animals received homologous stem cells injected at the ischemic myocardium area. A SPECT/CT evaluation showed decreased activity of the myocardial cells in the left ventricle one week post-infarction. Regeneration of the ischemic myocardium fifteen days post-infarction was recorded only in animals subjected to stem cell transplantation. These findings were also confirmed by histology and immunohistochemical analysis, with the significantly higher expression of GATA4 and Nkx2.5. Conclusions: The positive effect of mesenchymal stem cell transplantation in the ischemic myocardium was recorded. The application of SPECT-CT allowed a clear evaluation of both the quality and quantity of the living myocardium post-infarction, leading to a new approach in the research of cardiovascular diseases. From a clinical perspective, MSCT may be beneficial when accompanied by myocardial revascularization procedures.

## 1. Introduction

Cardiovascular diseases such as ischemic heart disease remain a major cause of death worldwide. Despite extensive research on pharmaceutical treatments, the mortality rates remain high, and, therefore, further investigations are needed [1,2,3,4]. The first study of the experimental induction of myocardial ischemia was reported in 1862. Since then, among the various techniques of inducing myocardial infarction in experimental models, the ligation of the left anterior descending artery (LAD) has become the most prominent [5,6,7].

Mesenchymal stem cells are multipotent cells that were first recognized in the bone marrow of adults. Adipose-derived stem cell (ADSC) transplantation is nowadays considered the ideal tissue choice due to the decreased possibility of rejection. They are characterized by neovascularization and the ability to differentiate to multiple cell lines, including adipocytes, osteoblasts, and chondroblasts. Furthermore, it is known that they can be differentiated to cardiomyocytes and can lead to newly formed cardiac tissue [7,8,9]. 

Despite numerous current experimental trials, there are still many questions unanswered, such as the type of cells to be used, the number of cells necessary to be transplanted, the technique and route of administration, and the confirmation of the improvement in the function of the target tissue, such as the myocardium [10,11].

The molecular markers used in our study to evaluate the action of ADSCs in the myocardium were GATA4, a zinc-finger transcription factor that is highly expressed in cardiomyocytes and is key to myocardial differentiation, and NKX2-5, a protein-coding gene that is involved in myoca32rdial conduction and contractility [8,9,12,13,14,15]. 

In vitro results showed an actively proliferating stem cell population 72 h after isolation, but it has been documented in immunochemical studies that the expression of stem cell markers needs approximately 7 days [16,17,18].

The goal of this study was to determine whether ADSC transplantation in the ischemic myocardium could cause the regeneration of the myocardial cells and increase the viable contractile tissue [1]. For this purpose, we used a previously described animal model with many prototype features [19,20,21]. Furthermore, to test heart function, we utilized a SPECT-CT evaluation before and after surgically induced ischemia [8,9,12]. As shown previously, this non-invasive method evaluates both anatomical and physiological changes, offers a serial quantitative approach to myocardial function and myocardial metabolism on a cellular level in the heart, and can be easily correlated with modern approaches to cardiovascular diseases, particularly when combined with molecular marker evaluations [22,23,24].

## 2. Methods

### 2.1. Animals and Study Protocol

Our animal model of inducing myocardial ischemia, as well as the documentation and monitoring of heart function as detected by SPECT-CT, has been previously described in detail [1]. The ADSCs were retrieved from the abdominal adipose tissue of 6 male rats (donors), while 40 female rats were used for the control, sham, and experimental groups, all of them weighing 280–350 g. This study was authorized by the national Animal Experiment Board of Greece and the Veterinary Association of Athens (license no.: 1870/23-04-2018) and was carried out in compliance with EU legislation relating to the conduct of animal experimentation [25]. The animals were hospitalized in a certified laboratory with the necessary equipment and veterinarian surveillance.

In accordance to the ARRIVE Guidelines and the 3Rs [26], animals were kept in groups of 3 per cage, with free access to water and food. The hygiene (waste, ventilation, etc.) and the pharmaceutical protocols were planned by the veterinarian committee according to the regulations of the authorized laboratory. After surgery, the animals were left to recover in cages alone or with other post-surgery animals in order to avoid cannibalism and other aggressive behaviors.

### 2.2. Surgical Procedure

Sterile conditions resembling those of an operating room in the surgical field were utilized. Accordingly, all instruments were sterilized in dry sterilization chambers. The surgical bed, a mattress specifically created for rodents and other small animals weighing below one kilogram, was connected to specialized equipment allowing continuous ECG recording, as well as an oximeter, thermometer, heater, and cooler. Nasopharyngeal intubation was achieved with a 17G tube connected to an inhalator with a mixture of sevoflurane 30–40%. Analgesia was achieved with the subcutaneous administration of Butorfanole (Dolorex) 10 mg/mL at the beginning of the procedure and every 3–5 h. Antibiotic coverage was achieved with the intramuscular administration of Oxytetracycline (Oxyvet 20%) 0.03 mL. A thermometer was placed in the rectal orifice of the animal, an oximeter was placed on the left foot, and the blood pressure was measured non-invasively by a cable placed on the left arm of the animal. A small thoracotomy was performed in the 4th intercostal space in order to enter the pericardium carefully. Except for the sham-operated group, all groups underwent an LAD ligation at its first third after the 1st diagonal branch using a 4-0 silk suture. The timeline of the experimental study is graphically presented in Figure 1. After 45 min of ischemia documented in the ECG by ST elevation, the injection of 0.4 mL of liquid, containing one million stem cells, took place in a cross pattern (0.1 mL per spot) around the ischemic area. Control animals (*n* = 10) received a buffer solution with heart-friendly electrolytes, while experimental animals (*n* = 10) received previously prepared ADSCs. Afterwards, the thoracotomy was closed per layer. At the end of the experiment (Figure 1), via a small laparotomy, the inferior vena cava was used for blood sampling, followed by the administration of excessive KCl resulting in myocardial arrest. Subsequently, the heart, lung, and liver were harvested and placed in formaldehyde. Cadavers were destroyed in biological waste material. There were 9 deaths: 2 deaths at the closure of the thoracotomy of the animal due to lung injury and pneumothorax, 2 deaths due to cardiac arrhythmias leading to ventricular fibrillation, 1 death from hyperthermia due to accidental removal of the thermometer, 1 death due to dextrocardia and the numerous manipulations for heart immobilization, 2 deaths in the first 24 h PO, and 1 due to cannibalism in the PO cage.

Drugs administered during the surgical procedure: (1) Tobrex eyedrops for eye protection were administered locally to the eyes. (2) Natural saline 0.9% and dextrose 5% for hydration were administered subcutaneously (10–20 mg/kg) at the beginning. (3) Dolorex (Boutorfanole 10 mg/mL) for analgesia was administered subcutaneously (1–2 mg/kg) as a starting dose and repeated every 3–5 h. (4) Oxyvet 20% (Oxytetracycline 200 mg/mL) for antibiotic coverage was administered intramuscularly (20 mg/kg) at the beginning and 3 days post-surgery.

Euthanasia: On the day of euthanasia (day 15), the animals underwent the same procedure of anesthesia as on the day of surgery (described earlier), followed by laparotomy. The identification and isolation of the inferior vena cava was performed, where the administration of 3 mL of KCl was established. In this way, bradycardia was achieved and finally the heart stopped in the diastole phase. Afterwards, tissue harvesting was performed—specifically, the removal of the heart, lungs, and liver which were placed in 10% neutral formaldehyde kits. The waste products were discharged and handled according to European legislation by the authorized staff of the experimental laboratory.

### 2.3. Isolation and Culture of ADSCs

ADSCs were isolated from 3-month-old male Wistar rats obtained from the experimental facility of the National Center for Scientific Research “Demokritos”, Athens, Greece, with official coding for the breeding and provision of animals (EL 25 BIO 019 and EL 25 BIO 020, respectively). The cells were collected from the subcutaneous layer of the adipose tissue of the abdominal wall and were washed with PBS, minced using two scalpels, and then digested in crude collagenase (1 mg/mL final concentration of collagenase; DMEM, Thermo Fisher Scientific, Inc., Waltham, MA, USA) for 30 min at 37 °C. Later, the material was centrifuged (200× *g* for 5 min) at 37 °C to discard the supernatant, and the pellet was resuspended in DMEM, 10% FBS (Thermo Fisher Scientific, Inc.), and 1% penicillin/streptomycin and then transferred to a culture flask. Overnight incubation at 37 °C was followed by the change of the medium in order to remove the nonadherent cells, and the attached cells were further cultured in the same medium, under standard culture conditions. Novel DNA synthesis was performed with dual labeling with 5-bromo-2′-deoxyuridine (BrdU) and 4′,6-diamino-2-phenylindole (DAPI) dihydrochloride (Sigma, Kawasaki City, Japan) [1,2,3]. Furthermore, ADSCs’ cell surface markers were examined, as described before [27].

### 2.4. SPECT-CT Acquisition and Reconstruction

Our SPECT-CT imaging technique was especially created for small rodents and the use of a specialized chamber allowed us to focus on the thoracic cavity of the animal; details have been published before [1]. Rats were injected with 200 μL of 1.5mCi-3mCi of 99mTc-Sestamibi, and imaging was performed at 20 min post-injection. SPECT-CT imaging was performed with the x-Cube and γ-Cube. CT acquisition and post-processing for the whole body was accomplished using a general-purpose protocol (50 kVp), performing an ISRA reconstruction with a 200 μm voxel size. All anatomical axes (short, long, and horizontal) were created for each animal, and 3D rendering with a color qualitative indication scale was used to display the myocardial activity at 7 and 14 days PO.

The measurement was performed in voxels from each cardiac chamber and the ratio of the left to right ventricle was evaluated in order to identify the changes in myocardial activity after establishing myocardial ischemia due to the surgical ligation of the LAD. See Figure 2.

### 2.5. Histological and Immunohistochemical Analysis

On day 15 PO, before euthanasia and under anesthesia, the heart was harvested and fixed in 10% neutral formaldehyde overnight and was routinely processed.

In our previous report of the experimental model, we documented histological proof of ischemia in the myocardium with a decrease in tissue fibers [1]. Furthermore, in this study, immunohistochemical staining for the analysis of antigen expression was performed. The expression of antigens was additionally assessed in 4-μm-thick tissue. Immunohistochemical staining allowed the visualization of antigens through the specific sequence of antibody–antigen bonding, followed by the bonding of a secondary antibody to the primary one and the creation of an enzymatic complex that causes a chromogenic reaction. The anti-rat antibodies were used in this study after specific steps of clearance. The molecular markers used in our study to evaluate the action of ADSCs in the myocardium were directed against GATA4 and NKX2.5, CD133, and CTGF (Table 1). We used polyclonal antibodies from the following kits: (1) Origene TA354470 for CD133, (2) Origene TA323092 for CTGF, (3) Origene AP20302PU-N for GATA4, and (4) Abcam ab214296. Diaminobenzidine was used as a chrome agent and light hematoxylin as a counter stain. Tissue sections, where the primary antibody was omitted, were used as negative controls.

### 2.6. Statistical Evaluation and Analysis

For quantitative variables, the selected data were expressed as the mean ± standard deviation (SD), while, for qualitative variables, they were expressed as frequencies and percentages. In order to analyze the normality of the quantitative variables, the Kolmogorov–Smirnov test was used. In order to compare the variables, both qualitative and quantitative pairwise comparisons between experimental groups were performed using one-way ANOVA with Bonferroni correction and the Chi-square test with Bonferroni correction, respectively. All tests were two-sided. A statistically significant difference was defined by a *p*-value < 0.05. Statistical analysis was performed using the statistical package SPSS version 21.00 (IBM Corporation, Somers, NY, USA).

## 3. Results

### 3.1. SPECT-CT Evaluation

The initial images were evaluated by performing a whole body spiral acquisition in a high-resolution protocol. The visualization was firstly standardized by scanning all the animals before surgery, and the distribution of the radioactive tracer at 7 and 14 days after the surgical ligation of the LAD proved the changes related to ischemia (Figure 2). The differences between the healthy and ischemic myocardial tissue were easily recognized after the axial and 3D reconstruction, leading to the creation of a physiological and anatomical map of the rat’s heart (Figure 2). In the healthy myocardial cells, the radioactive absorbance was high as the cells were highly metabolic, while, in the ischemic cells, the absorbance was decreased or absent (Figure 3, left column). It is notable that, in the animals receiving stem cell transplantation, the SPECT-CT evaluation revealed definite signs of regeneration and an increase in myocardial activity (Figure 3, right column; Figure 4).

### 3.2. Left Ventricle/Right Ventricle Ratio (LV/RV Ratio)

The creation of the ratio of the left ventricular area to the right ventricular area (LV/RV), which is purely mathematical and not influenced by any other parameters of the experimental method, such as the animal weight, the dosage of the radioactive substance, or the absolute number of voxels, led to a clear analysis of the distribution of the radioactive particle. The results are summarized in Table 2. Due to the ligation of the LAD, the induced ischemia causes a decreased left ventricular area; therefore, the LV/RV ratio changes pre- and postoperatively. In more detail, healthy individuals presented an LV/RV ratio of about 8.7 ± 0.3 (day −7) 7 days prior to surgery, and this remained stable in the animals that underwent only thoracotomy without ligation. However, there was a statistically significant difference between groups with respect to the ratio of variables at 7 days (*p* < 0.005), and pairwise comparisons showed that the sham-operated animal group presented higher values compared to the control group (*p* < 0.005). The animals that belonged to the control group had also a significant decrease in the activity of the myocardium, with an LV/RV ratio of 7.5 ± 0.2 post-surgically. The animals in the sham-operated group showed no increase in myocardial activity fourteen days post-surgery, while the animals in the experimental group that received the mesenchymal stem cells showed an increase in functioning myocardial cells and specifically the values of the LV/RV ratio were 8.00 ± 0.2, demonstrating thus the beneficial effect of stem cell transplantation, as clearly depicted in Figure 5.

### 3.3. Immunohistochemical Analysis and Evaluation of Immunohistochemistry

Strong immunostaining in myocardial cells was considered positive. The comparison and results are summarized in Table 2 and the immunohistochemical analysis results are illustrated in Figure 6a,b. There was a statistically significant difference between groups regarding the percentage of positive results for the CTGF variable (*p* = 0.011), as well as with the CD133 variable. Pairwise comparisons showed that the experimental group that received the stem cells presented a higher percentage of positive results for the CD133 variable compared to the sham-operated (*p* < 0.005) and control groups (*p* < 0.005), respectively. There was also a statistically significant difference between groups with respect to the percentage of positive results for the GATA4 variable (*p* < 0.005). Pairwise comparisons showed that the experimental group (stem cells) presented a higher percentage of positive results for the GATA4 variable compared to the sham-operated (*p* < 0.005) and control groups (*p* < 0.005), respectively. There was a statistically significant difference between groups with respect to the percentage of positive results for the Nkx2.5 variable (*p* < 0.005). Pairwise comparisons showed that the experimental group presented a higher percentage of positive results for the Nkx2.5 variable compared to the sham-operated (*p* = 0.045) and control groups (*p* < 0.005), respectively. These favorable findings indicate the beneficial effect of SCT in this rat model, in terms of the regeneration of the ischemic myocardium (Figure 6a,b).

## 4. Discussion

Ischemic heart disease is a leading cause of death worldwide; thus, quick and accurate diagnosis and effective therapeutic protocols are needed [1,2,3,4,5,6,7]. 

Numerous experimental protocols have been created in order to understand the mechanism of myocardial regeneration and develop guidelines for pre-clinical and clinical trials [10,11].

The overall goal of such experimentation is the understanding of the ischemic mechanism in such a way that the approach of regeneration is directed via the fastest and most effective pathway. The experimental model used in our study sought to translate the clinical symptoms of myocardial infarction into a qualitative and quantitative presentation, while the therapy with stem cells served to ensure long-term regeneration, clearly recognizable in vivo and postmortem [2,3,4,5,6].

Therefore, we strongly believe that our findings can be a helpful tool in the hands of researchers worldwide in order to understand, evaluate, present, and cure the ischemic conditions of the myocardium [19,21].

This study is important for two major reasons. The first is the ability to evaluate experimentally the degree of ischemia in living animals, instead of the commonly used method of postmortem analysis [8,9,12,13,14,15]. 

Secondly, the use of specific heart-oriented mesenchymal stem cells is leading to a new era of experimentation that is organ-target-specific and can take stem cell transplantation towards a new level of therapeutical goals [16,17,18].

The experimental model (rodents) was chosen due to its high compatibility with the human physiology and anatomy concerning the heart [1]. Therefore, the knowledge gained can be compared and replicated worldwide with great accuracy for in vivo evaluation [8]. The establishment of myocardial ischemia and infarction by ligation of the LAD is highly demanding due to increased mortality rates resulting from the stress of animals and the waste products of the ischemic cells in circulation [19,20,21]. 

Through the standardization of the technique via a left thoracotomy, we managed to decrease the mortality rate to less than 20%, making it attractive for in vivo experimentation and real-time results, in contrast to the use of pharmaceutical causes of infarction, which only resemble an actual heart attack [10,11]. 

Furthermore, the SPECT-CT imaging of the living animals pre- and post-infarction allowed us to understand and present the changes that occur in a living myocardium both physiologically and anatomically, with measurements that are of high qualitative and quantitative value. The use of SPECT creates a very recognizable pattern of living cells absorbing the radioactive substances, which can be measured and evaluated [28,29,30,31]. The CT gives the exact anatomical structure of the heart and the changes in the left ventricle because of the infarction. Furthermore, the regeneration and the increase in contractility of the heart tissue in the animals that received the stem cell therapy was undeniable [22,23,24].

Mesenchymal stem cells have been intensively studied in the past few years, in an effort to find new therapeutical methods for numerous diseases [8,9,12,13,14,15]. 

Their use in myocardial tissue is not yet popular since there have not been any particular cells addressed to this tissue with comparable results. Our goal was to isolate myocardial-specific cells, specifically GATA4 and Nkx2.5, and to study their specific action in the post-infarcted myocardium [16,17,18]. Most researches have been using cell cultures with positive results, but it is challenging to study their efficacy in living animals. In this study, the administration of the cells was performed directly in the infarcted area in the beating heart and the positive regenerative results were confirmed via the immunohistochemical analysis of the heart tissue. It was confirmed that CD133 was present only in the animals of the experimental group as a general stem cell marker, while CTGF was present in both the experimental and control groups as a general regenerative marker, but not in the sham-operated group, where no ischemia was induced. It was proven that the animals that received the stem cells had an increased volume of the left ventricle and better contractility at 15 days PO, as shown in Table 3 and Figure 5. Only in the experimental group of animals, the stem cells were isolated and recognized postmortem. It is notable that, using this model and methodology, the great challenges of such an experiment were addressed in a smooth and effective manner.

## 5. Limitations of the Study

Among the limitations of this study are as follows. (1) Only female rats were used as stem cell receivers and only male rats were used as stem cell donors. (2) The period of survival and monitoring until euthanasia was short. This was necessary for reasons related to the obtained license for this experiment. (3) The weight of the animals was not the same for all, but was within a range corresponding to adult rats of the same age; similarly, the weight of their hearts was within an acceptable range.

## 6. Conclusions

In conclusion, this novel, valid, and highly reproducible infraction model and the methodology used gave a transparent vision of the myocardial activity, with both qualitative and quantitative parameters. Furthermore, verification by immunohistochemical analysis demonstrated the therapeutic potential of ADSC transplantation. From a clinical perspective, therefore, concurrent ADSC administration may be applied in combination with coronary artery bypass grafting. However, additional studies are required to further clarify the regenerative effect of stem cell transplantation in the ischemic myocardium.

## Figures and Tables

**Figure 1 diagnostics-14-00401-f001:**
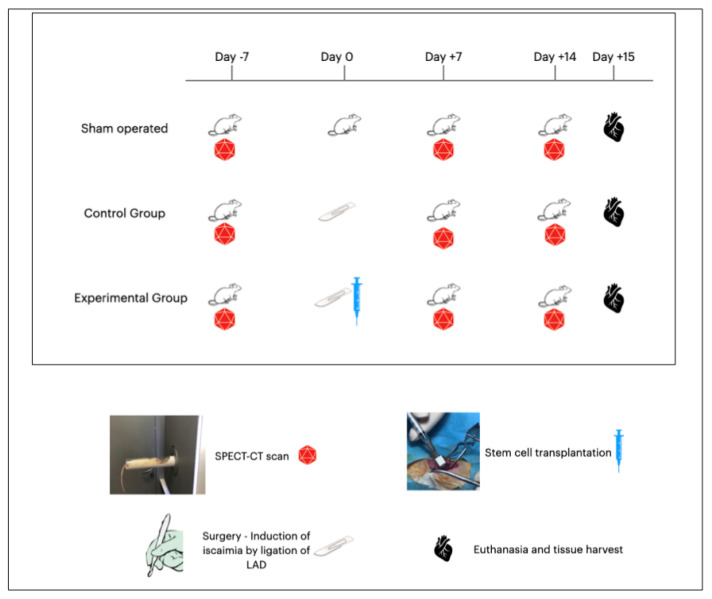
Timeline of the experimental study. All animals underwent SPECT-CT scan 7 days prior to surgery. Day 0 was the day of surgery when sham-operated animals underwent left thoracotomy without any further manipulation, control animals underwent left thoracotomy and surgically induced ischemia via ligation of LAD, and experimental animals underwent left thoracotomy and ligation of LAD followed by installation of stem cells at the point of ischemia in a cross pattern. All animals underwent SPECT-CT 7 days after surgery and 15 days after surgery. Finally, on the 15th day post-surgery, euthanasia took place, followed by tissue harvesting.

**Figure 2 diagnostics-14-00401-f002:**
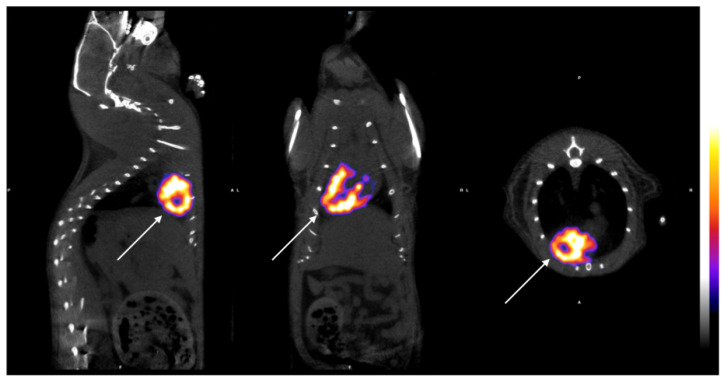
Images of the left ventricle (arrows). SPECT/CT imaging of all axes’ views (longitudinal, sagittal, and transverse from left to right) in healthy myocardium. SPECT-CT images were collected to visualize the myocardial functionality. The color bar indicates the difference in accumulated radiopharmaceutical, directly linked to the level of blood circulation (deep blue being the lowest and white the highest). The color bar indicates a qualitative estimation of myocardial functionality, while quantitative values were extracted from the voxels of each cardiac chamber.

**Figure 3 diagnostics-14-00401-f003:**
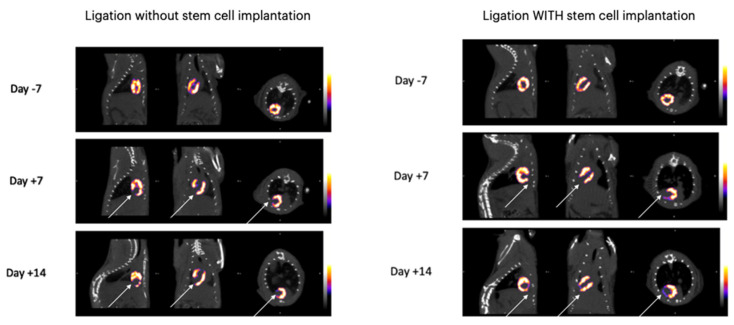
SPECT/CT imaging of the experimental group showing the regeneration of the myocardium (lowest values shown by deep blue and high values by white).

**Figure 4 diagnostics-14-00401-f004:**
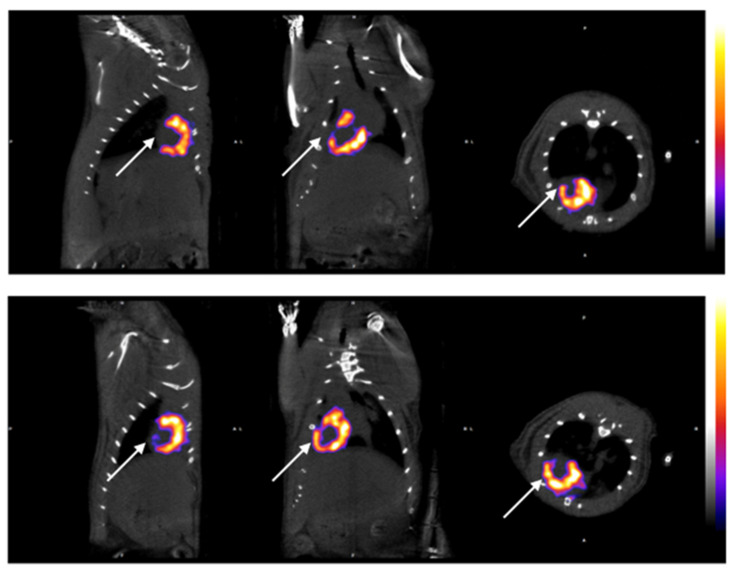
Myocardial activity 7 and 14 days after stem cell administration in post-infarcted area. The color bar indicates the different levels of the accumulated radiopharmaceutical, directly linked to the level of blood circulation (lowest values shown by deep blue and high values by white). All results were measured in voxels from each cardiac chamber. Photo above shows the decreased absorption of radioactive particles in the apex of the heart supplied by LAD due to surgically induced ischaimia. (Day +7) Photo below shows increased absorption of radioactive substance 14 days post stem cell administration proving the presence of myocardial regeneration in prior ischemic areas of the heart apex. These animals with stem cell implantation belong to the experimental group (*n* = 10) and received 1,000,000 cells.

**Figure 5 diagnostics-14-00401-f005:**
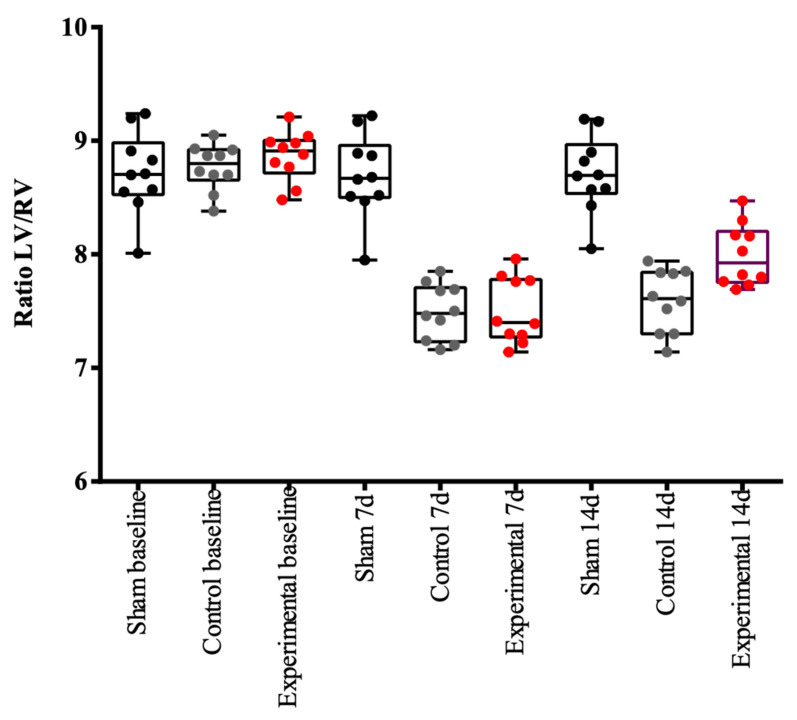
Left ventricle to right ventricle ratio (LV/RV) changes among sham, control, and experimental groups of animals during the period of experimentation. During the period of experimentation, the ratio of the myocardial volume of the left to right ventricles changed according to the presence and the degree of ischemia. In order to compare the variables, both qualitative and quantitative pairwise comparisons between experimental groups were performed using one-way ANOVA with Bonferroni correction and the Chi-square test with Bonferroni correction, respectively. All tests were two-sided. Statistically significant differences were defined by a *p*-value < 0.05. Black dots represent the sham-operated group, where no change was seen during this time—the LV/RV ratio was about 8.7 ± 0.3 at every measurement (days −7, +7, +14). Grey dots represent the control group, where the LV/RV ratio decreased after the ligation of the LAD and remained low during the period after LAD ligation and establishment of ischemia—the LV/RV ratio was about 7.5 ± 0.27 (days +7, +14). Red dots represent the experimental group, where the LV/RV ratio decreased after ligation of the LAD—LV/RV ratio was about 7.59 ± 0.28 (day +7). It tended to return to normal on day +14 following stem cell administration—LV/RV ratio was about 8 ± 0.27 (Day +14).

**Figure 6 diagnostics-14-00401-f006:**
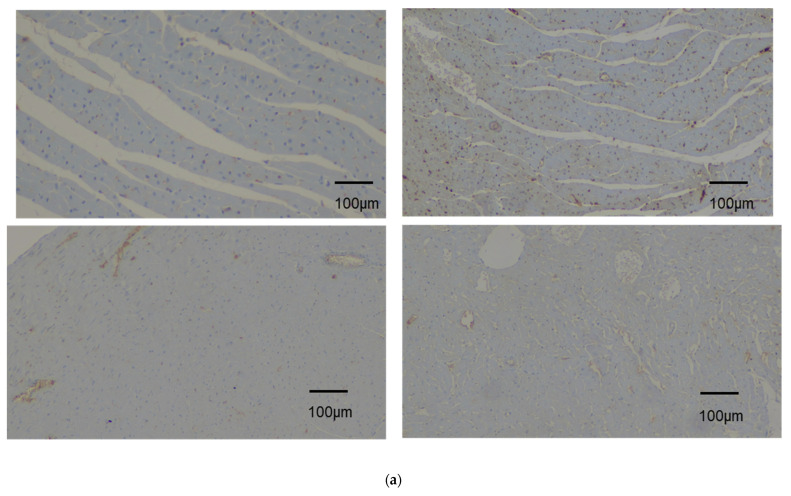
(**a**) Immunohistochemical analysis of myocardium before stem cell administration. (**b**) Following stem cell administration, immunohistochemical analysis of four factors—antigens in myocardium without stem cell administration (control and sham-operated animal groups). All photos were taken with 10× magnification. Top left: CD133 antigen negatively stained. Top right: CTGF antigen mildly stained positively. Bottom left: GATA4 antigen negatively stained. Bottom right: Nkx2.5 antigen negatively stained. Immunohistochemical analysis of four factors—antigens in myocardium after stem cell administration. All photos illustrated were taken with 10× magnification. From top to bottom: (1) CD133 antigen positively stained, (2) CTGF antigen positively stained, (3) GATA4 antigen positively stained, (4) Nkx2.5 antigen positively stained.

**Table 1 diagnostics-14-00401-t001:** Immunohistochemical markers used for antigen–antibody recognition of stem cell administration.

Molecular Factors	Brand Kit Used	Properties	Dilution
GATA-4	Origene AP2030PU-N	Key role in cardiac development	Supplier instruction: IHC-P 1/50-1/200 Used dilution: IHC-P 1/75
Nkx2.5	Abcam Ab214296	Differentiation of myocardial lineage	Supplier instruction: IHC-P 1/100-1/500 Used dilution: IHC-P 1/300
CD133	Origene TA354470	Cell differentiation, proliferation, and apoptosis	Supplier instruction: IHC 2-10 μg/mL Used dilution: IHC 6 μg/ml
CTGF	Origene TA323092	Connective tissue mitoattractant secreted by vascular endothelial cells	Supplier instruction: IHC-Fr 1/200 Used dilution: IHC-Fr 1/200

**Table 2 diagnostics-14-00401-t002:** Statistical analysis of four stem-cell-related factors between the experimental groups. There was a statistically significant difference between groups with respect to percentage of positive results for the CD133, GATA4, and Nkx2.5 variables (*p* < 0.005). Pairwise comparisons showed that the experimental group presented higher percentages of positive results for the CD133, GATA4, and Nkx2.5 variables compared to the sham (*p* = 0.045) and control (*p* < 0.005) groups, respectively. Pairwise comparisons between groups were performed using one-way ANOVA with Bonferroni correction.

Variable	Group			*p*-Value
	Sham	Control	Stem Cells	
CTGF negative/positive; *n* (%)	6 (60)/4 (40) ^†^	11 (100)/0 (0) **	4 (40)/6 (60)	0.011
CD133 negative/positive; *n* (%)	10 (100)/0 (0) **	11 (100)/0 (0) **	2 (20)/8 (80)	<0.005
GATA4 negative/positive; *n* (%)	9 (90)/1 (10) **	11 (100)/0 (0) **	0 (0)/10 (100)	<0.005
Nkx2.5 negative/positive; *n* (%)	8 (80)/2 (20) *	11 (100)/0 (0) **	2 (20)/8 (80)	<0.005

* *p* < 0.05 vs. stem cells, ** *p* < 0.005 vs. stem cells, ^†^
*p* < 0.05 vs. control.

**Table 3 diagnostics-14-00401-t003:** Comparison of LV/RV area ratio between the animal groups at the specific time points of experimentation. In healthy individuals, the LV/RV ratio 7 days prior to surgery is about 8.7 ± 0.3 (day −7) and remains stable in the sham-operated animal group. There is a statistically significant difference between groups with respect to the ratio of variables at day 7.

Variable	Group			*p*-Value
	Sham	Control	Stem Cells	
RATIO LV/RV (−7d)	8.72 ± 0.36	8.77 ± 0.20	8.86 ± 0.22	0.480
RATIO LV/RV (7d)	8.69 ± 0.37	7.50 ± 0.24 *	7.50 ± 0.29 *	<0.005
RATIO LV/RV (14d)	8.71 ± 0.34	7.59 ± 0.28 *	8.00 ± 0.27 *,^†^	<0.005

* *p* < 0.005 vs. sham, ^†^
*p* < 0.05 vs. control.

## Data Availability

Data are available upon request.

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
