# Peer review of "Mesenchymal Stem Cell Transplantation Has a Regenerative Effect in Ischemic Myocardium: An Experimental Rat Model Evaluated by SPECT-CT Assessment"

_diagnostics, 2024, doi:10.3390/diagnostics14040401_

Round 1

Reviewer 1 Report

Comments and Suggestions for Authors

The authors describe the use of a novel rat model of ischemic heart disease and have used adipose tissue derived- MSCs for the regeneration of heart tissue after ischemia. SPECT-CT has been used to evaluate the output of the cell therapy.

The subject is of outmost importance for regenerative medicine field, considering that heart disease is the leading cause of death worldwide.

Several things must be clarified and corrected before publication of the manuscript:

1.       In the abstract, the definition of CABG is missing.

2.       In the introduction, the authors state “mesenchymal stem cells are pluripotent cells…”. This is not right and must be corrected, once MSCs are multipotent cells.

3.       In methods section, the number of MSCs and a justification of the number of cells choose must be indicated.

4.       The quality of the figure 1 should be improved. Please use graphic art editors.

5.       In “Drugs administered during the surgical procedure”, authors state several times “at the beginning”. Please be more precise.

6.       In figure 2, please indicate more details on the anatomical structures of rats’ heart.

7.       In section 2.5, a list of antibodies, catalog number, brand, and dilution used should be presented in a table.

8.       The data presented in table 1 should be included in section 3, not in methods section. In addition, full explanation on how the quantifications presented in table 1 were made should be provided.

9.       Section 3.2 starts with “Moreover, the creation of the left ventricular to right ventricular…”. It seems that something is missing. Please reformulate the sentence.

10.   Please explain why ischemia due to LAD causes decrease in left ventricular area.

11.   In section 3.3, please explain the choice of the specific markers in the immunostaining and what is the importance of the increase in these markers after MSC administration.

12.   Figure 6: Quality of images is very poor. Please provide higher resolution and higher magnification images.

13.   Immunostaining should be made to show if the MSC are present in the injured area 7 and 15 PO. Discussion on whether the beneficial effects observed after MSC administration are due to the cells or their secreted factors should be made.

Comments on the Quality of English Language

The overall quality of English must be improved. There are several sentences with confused ideas, so better organization of ideas and the overall manuscript  should be taken in consideration.

Author Response

Thank you for your helpful and useful comments. We have the pleasure to submit our revised manuscript for consideraation for publication in the journal. We hope we have managed to address all points of criticism and all the comments and suggestions made by you to the attached file we upload.  Looking forward to your feedback.

Reviewer 2 Report

Comments and Suggestions for Authors

Method

Stemness of MSCs are not confirmed.

Result

Table 1 and 2 titles are not clear. Astric and cross is not defined and p-value seems to be wrong for such differences. Please report exact p-value.

Figure 5 seems have some signifacnt differences between the times. Please show them by lines.

Figure 6 is very low quality.

Figures need to have scale bar instead of magnifications.

Please seperate conclusion from limitation.

Author Response

Thank you for your helpful and useful comments. We have the pleasure to submit our revised manuscript for consideraation for publication in the journal. We hope we have managed to address all points of criticism and all the comments and suggestions made by you in the attached file we upload.  Looking forward to your feedback.

Round 2

Reviewer 1 Report

Comments and Suggestions for Authors

The reviewer appreciate the effort done improving the manuscript.

There are still some minor revisions to be done:

1) In table 1, when is read "defferentiation" should be read "differentiation".

2) Still in table 1, it was asked by the reviewer to include the dilution used in the present study, not the predicted dilution recomended by the supplier.

Author Response

Thank you very much for your recognition of our improvements and also for your helpful guidance and comments! You help us very much improving our manuscript!

Comment 1 : In table 1, when is read "defferentiation" should be read “differentiation".

Answer : Thank you very much for noticing this. Appropriate change is done in the table 1.

Change : correction in table 1

Comment 2 :  Still in table 1, it was asked by the reviewer to include the dilution used in the present study, not the predicted dilution recomended by the supplier.

Answer : Thank you very much for asking this specifically. The instructions by the supplier were general and we used the suitable dilution for our samples by our laboratory experience. 

Change : Used dilutions are now mentioned in the table 1. 

Reviewer 2 Report

Comments and Suggestions for Authors

Thanks for the corrections made by the respected authors. But two corrections are still needed. Regarding the box-whisker chart, the statistical differences between the desired groups should be analyzed with a non parametric indipendent or paired test and these differences should be displayed with the help of the lines above the boxes. Also, in histology images, it is necessary to use the scale bar to show the magnification, and it is not correct to write the magnification in the caption of the photo. Because the size of the photos will change during the editing process.

Author Response

Thank you very much for your kind comments and suggested corrections. You are truly helping us improve our manuscript. 

Comment 1 :  Regarding the box-whisker chart, the statistical differences between the desired groups should be analyzed with a non parametric indipendent or paired test and these differences should be displayed with the help of the lines above the boxes. 

Answer : Thank you very much for noticing this in the chart. Statistical details explaining the analysis is made in the main text and in table 3.

“ 2.6. Statistic evaluation and analysis

For quantitative variables the selected data were expressed as mean ± standard deviation (SD) while for qualitative variables as frequencies and percentages. In order to analyze the normality of the quantitive variables the Kolmogorov—Smirnov test was used. In order to compare the variables both qualitative and quantitative pairwise comparisons between experimental groups was performed using one way ANOVA with Bonferroni correction and the Chi-square test with Bonferroni correction, respectively. All tests were two-sided. Statistically significant difference was defined by a   p-Value < 0.05. Statistical analysis was performed using the statistical package SPSS version 21.00 (IBM Corporation, Somers, NY, USA). “

Change : Extra information added in the chart accordingly to your comments. 

Comment 2 : In histology images, it is necessary to use the scale bar to show the magnification, and it is not correct to write the magnification in the caption of the photo. Because the size of the photos will change during the editing process.

Answer : Thank you very much for explaining the necessity of the scale bar. It is improving the visualization of the manuscript. 

Change : Scale bar added in images of immunohistochemistry in the manuscript. (Figure 6)